# Rapid Detection of *Salmonella* Enteritidis, Typhimurium, and Thompson by Specific Peak Analysis Using Matrix-Assisted Laser Desorption Ionization Time-of-Flight Mass Spectrometry

**DOI:** 10.3390/foods10050933

**Published:** 2021-04-23

**Authors:** Seung-Min Yang, Eiseul Kim, Dayoung Kim, Jiwon Baek, Hyunjin Yoon, Hae-Yeong Kim

**Affiliations:** 1Institute of Life Sciences & Resources and Department of Food Science and Biotechnology, Kyung Hee University, Yongin 17104, Korea; ysm9284@gmail.com (S.-M.Y.); eskim89@khu.ac.kr (E.K.); ekdudvv4589@naver.com (D.K.); 2Department of Molecular Science and Technology, Ajou University, Suwon 16499, Korea; qorwldnjs78@ajou.ac.kr (J.B.); yoonh@ajou.ac.kr (H.Y.)

**Keywords:** *Salmonella* serotyping, MALDI–TOF MS, identification, detection, specific peak, Enteritidis, Typhimurium, Thompson

## Abstract

Rapid detection of *Salmonella* serovars is important for the effective control and monitoring of food industries. In this study, we evaluate the application of matrix-assisted laser desorption/ionization time-of-flight mass spectrometry for the rapid detection of three serovars, Enteritidis, Typhimurium, and Thompson, that are epidemiologically important in Korea. All strains were identified at the genus level, with a mean score of 2.319 using the BioTyper database, and their protein patterns were confirmed to be similar by principal component analysis and main spectrum profile dendrograms. Specific peaks for the three serovars were identified by analyzing 65 reference strains representing 56 different serovars. Specific mass peaks at 3018 ± 1 and 6037 ± 1, 7184 ± 1, and 4925 ± 1 *m/z* were uniquely found in the reference strains of serovars Enteritidis, Typhimurium, and Thompson, respectively, and they showed that the three serovars can be differentiated from each other and 53 other serovars. We verified the reproducibility of these mass peaks in 132 isolates, and serovar classification was achieved with 100% accuracy when compared with conventional serotyping through antisera agglutination. Our method can rapidly detect a large number of strains; hence, it will be useful for the high-throughput screening of *Salmonella* serovars.

## 1. Introduction

*Salmonella*, one of the main causes of foodborne diseases, is a significant public health concern. *Salmonella* is known to be spread primarily through the consumption of contaminated food or water [1,2,3]. *Salmonella* infection can cause serious illness in humans, especially infants and the elderly, causing diseases such as typhoid fever and gastroenteritis and even leading to death [4]. It is estimated that about 1.2 million cases of *Salmonella* infection occur annually in the United States alone, and *Salmonella* has the second-highest rate of incidence among the pathogenic bacteria that cause food poisoning in Korea [5,6]. The genus *Salmonella* is divided into two species: *S. bongori* and *S. enterica*. *S. enterica* is further subdivided into six subspecies [7]. Of these, *S. enterica* subsp. *enterica* is of clinical relevance and is the most frequently isolated subspecies [7,8]. *Salmonella* serotyping according to the White–Kauffmann–Le Minor scheme is based on a combination of the biochemical identification of somatic O, flagella H, and capsular Vi antigens and is recognized worldwide as a gold standard for differentiation below the subspecies level [7,9].

Although more than 2600 serovars exist, Enteritidis and Typhimurium are the major serovars of foodborne infection in humans and are considered very important in public health [10]. Serovar Enteritidis is the most common serovar of infection of humans worldwide [11]. According to the World Health Organization, serovar Enteritidis is the most common cause of gastroenteritis, although previously, serovar Typhimurium was the main cause [10]. Recently, a large-scale outbreak of gastroenteritis caused by serovar Thompson was reported in Korea, with a total of 2207 people infected [12]. Epidemiological and traceback evidence showed that egg whites used to make a chocolate cake were infected with Thompson, and the chocolate cake was announced as the source of infection [12]. In order to ensure the safety of food, regulations on the microbial inspection standards of edible eggs were revised and adjusted in 2019 so that Enteritidis, Typhimurium, and Thompson are not found in food. Thus, timely detection of the three prominent serovars is very important.

Detection of the causative pathogen is usually an essential step in epidemiological investigation [4]. Unfortunately, serotyping, typically performed by slide agglutination, is time-intensive, costly, complicated, and laborious and requires more than 150 specific antisera [7,13]. This method also requires that workers have the expertise necessary to interpret the aggregation results. Biochemical tests and morphological descriptions may produce ambiguous results [4]. Although whole-genome sequencing is becoming ever more accessible and is useful for serotyping because it provides comprehensive genetic information, it is not practical for sequencing multiple isolates simultaneously [4]. However, matrix-assisted laser desorption ionization time-of-flight mass spectrometry (MALDI–TOF MS) is a powerful tool that can detect bacterial strains rapidly and accurately. This technique is increasingly being used for the detection of bacterial strains [14,15,16,17,18]. While most of these studies have demonstrated the usefulness of the technique for species or subspecies identification, bacterial detection below the subspecies level, such as the discrimination of serovars, has rarely been addressed [7].

In this study, we evaluate the performance of MALDI–TOF MS for the detection of major *Salmonella* serovars isolated from various food samples, including egg white, and discover peaks specific to serovars Enteritidis, Typhimurium, and Thompson by analyzing the mass spectra of *Salmonella* serovars. The accuracy of specific peaks from the MALDI–TOF MS spectra was confirmed using conventional serotyping through antisera agglutination.

## 2. Materials and Methods

### 2.1. Isolation of Salmonella Strains

Reference strains of 12 Enteritidis, Typhimurium, and Thompson *Salmonella* serovars and 53 other serovars used in this study are presented in Table 1. *Salmonella* strains were isolated from various sources, such as processed food, fresh food, chicken meat, vegetables, egg white, and livestock. These strains were isolated as recommended by the FDA [19]. One milliliter of sample filtrate was pre-enriched by inoculation into 9 mL of buffered peptone water (Difco, Becton & Dickinson, Sparks, MD, USA), and the culture was incubated at 37 °C for 18 h. The cultured cells were inoculated into 10 mL of Rappaport–Vassiliadis broth (Difco) and incubated at 42 °C for 24 h. The inoculum was streaked on xylose lysine desoxycholate agar (XLD, Difco) and incubated at 37 °C for 24 h under aerobic conditions. Then, a colony suspected of being *Salmonella* was screened on XLD agar. All reference strains and isolates were cultured on tryptic soy agar (Difco) at 37 °C for 18 h under aerobic conditions and then subcultured in tryptic soy broth for storage in 30% glycerol (*v/v*) at −80 °C.

### 2.2. Identification of Serovar-Specific Peaks Using MALDI–TOF MS

Proteins of *Salmonella* strains were extracted using an ethanol/formic acid extraction method [14,15]. A loopful of *Salmonella* cells was suspended in 300 µL deionized water, and 900 µL of ethanol was added. The cell suspension was centrifuged at 13,600× *g* for 5 min, and the pellet was dried at room temperature. The dried pellet was resuspended in 50 µL of 70% formic acid and 50 µL of acetonitrile and centrifugation at 13,600× *g* for 5 min. Then, 1 µL of supernatant was spotted onto an MSP 96 polished steel target plate (Bruker Daltonics, Bremen, Germany) and allowed to dry at room temperature. After drying, each spot was overlaid with 1 µL of matrix solution (α-cyano-4-hydroxycinnamic acid in 50% acetonitrile and 2.5% trifluoroacetic acid). The acquisition of mass spectra was performed using a Microflex LT bench-top mass spectrometer (Bruker Daltonics). The mass spectra were measured using FlexControl software, with the default parameter settings, for bacterial identification. Calibration of the instrument was performed using a bacterial test standard (Bruker Daltonics) consisting of *Escherichia coli* protein extract. The obtained mass spectra were analyzed for the identification of serovar-specific peaks, as identified in previous studies [16,20,21]. Before identifying the specific peaks, the raw spectra were preprocessed by normalizing, smoothing, and baseline subtraction [22]. Spectra of poor quality, such as peaks with significant background noise or very low intensity, were excluded. Serovar-specific peaks were determined by comparing the spectra for different serovars using FlexAnalysis software version 3.4 (Bruker Daltonics).

### 2.3. Identification of Serovars by BioTyper and Specific Peaks

For the detection of *Salmonella* isolates, proteins of strains were extracted using a direct transfer method [15]. A fresh colony was smeared on an MSP 96 polished steel target plate (Bruker Daltonics) using a toothpick and overlaid with 1 µL matrix solution. After drying, the target plate was loaded into a Microflex LT bench-top mass spectrometer, which includes BioTyper database version 3.4 (5627 database entries), under the same conditions described above. The results were presented as a score between 0 to 3. The score interpretation of the four categories was as follows: a score of ≥2.3 indicates a high level of probability of species, scores between 2.000–2.299 indicate probable species identification, scores between 1.700–1.999 indicate probable genus identification, and a score < 1.7 indicates no reliable identification. To identify serovars with specific peaks, raw spectra were normalized, and the peak intensity and areas of the isolates were obtained using FlexAnalysis software (Bruker Daltonics). *Salmonella* serovars were determined by identifying the presence or absence of serovar-specific peaks. Main spectrum profile (MSP) dendrograms and principal component analysis (PCA) for all strains were conducted as per the standard operating procedure using MALDI BioTyper software version 3.1 (Bruker Daltonics). The mean peak masses, peak frequencies, and peak intensities were visualized.

### 2.4. Serotyping through the Agglutination of Antisera

The serotypes of the isolates used in this study were confirmed using a traditional serotyping method, as described in a previous study [1], and compared with the results of MALDI–TOF MS. Serology was performed using slide agglutination tests with commercial monovalent or polyvalent somatic O and flagellar H antisera (Difco), according to the manufacturer’s instructions. Serovars of isolates were determined using antigenic formulae based on the White–Kauffmann–Le Minor scheme [9].

## 3. Results

### 3.1. Isolation and Identification of Salmonella Strains

In the food and livestock samples, presumptive *Salmonella* colonies were grown on XLD agar and identified using MALDI–TOF MS with the BioTyper database. One hundred and thirty-two colonies were identified as *Salmonella* species. *Salmonella* was isolated from processed food (31 strains), fresh food (19 strains), chicken meat (45 strains), vegetables (18 strains), egg white (2 strains), and livestock (17 strains) (Table 2). Of these isolates, 88 strains (66.67%) were identified at the highly probable species level (log score ≥ 2.3) and 44 strains (33.33%) were identified at the probable species level (log score of 2.0–2.3). All isolates were identified as *Salmonella* species at the genus level by MALDI–TOF MS with the BioTyper database.

In total, 12 reference strains and 132 isolates were analyzed by MALDI–TOF MS to evaluate the robustness of the BioTyper database. MSP dendrograms and PCA clustering are useful for distinguishing between closely related strains and determining associations between the strains [22]. These methods were used to confirm the discriminative power of mass peak analysis for serovar detection. All strains of serovars Enteritidis, Typhimurium, and Thompson fell into three distinct groups in the dendrogram (Figure 1). The first cluster contained serovar Enteritidis, the second cluster contained Typhimurium, and the third cluster included all three serovars—Enteritidis, Typhimurium, and Thompson. PCA was used to cluster specimens according to their relative intensities and mass values (Figure 2). This clustering showed that the three serovars were not clearly separated. Additionally, a difference in mass peak pattern according to the source of isolation was not observed. Therefore, both the MSP dendrogram and PCA confirmed that the mass spectra of the three serovars were similar.

### 3.2. Analysis of Serovar-Specific Peaks

The mass spectrum profiles of *Salmonella* Enteritidis, Typhimurium, and Thompson showed similar patterns (Figure 3). The discriminative ability at the serovar level was evaluated by analyzing multiple mass spectra obtained from 53 different serovars. A total of 499 mass peaks were extracted from the spectrum profiles of 12 reference strains of serovars Enteritidis, Typhimurium, and Thompson and analyzed for peak value and intensity. In this process, one peak was excluded due to poor quality spectra with low intensity (36.189 arbitrary units). The peaks from these serovars were compared with 770 peaks extracted from 53 other serovar strains to confirm that they were unique peaks not found in other serovars.

In *Salmonella* Enteritidis, mass peaks at 3018 ± 1 and 6037 ± 1 *m/z* were found in all Enteritidis strains; these peaks were present in all six Enteritidis strains tested but absent in other *Salmonella* serovars, including Typhimurium and Thompson (Table 3). A total of 23 peaks were found in all Typhimurium strains; Typhimurium was characterized by the presence of a mass peak at 7184 ± 1 *m/z*. *Salmonella* Thompson could be clearly detected by the presence of a mass peak at 4925 ± 1 *m/z*, which was not found in the other 55 serovars. Mass peaks at 3018 ± 1, 6037 ± 1, 7184 ± 1, and 4925 ± 1 *m/z* were unique to serovars Enteritidis, Typhimurium, and Thompson and 20 other pathogenic bacterial strains (Table 4).

### 3.3. Identification of Isolates by Serovar-Specific Peaks

To validate our approach to the detection of Enteritidis, Typhimurium, and Thompson, the most epidemiologically important serovars, 132 *Salmonella* isolates were analyzed using MALDI–TOF MS. Serovar-specific peaks for *Salmonella* Enteritidis, Typhimurium, and Thompson were identified by the mass peaks described above (Figure 4). A total of 132 isolates were identified using these mass peaks to confirm that these peaks, identified in the reference strains, were present in many isolates. In total, 55 isolates were identified as serovar Enteritidis by peak analysis (Table 5), while 74 isolates were identified as serovar Typhimurium and 3 as Thompson. The mass peak at 3018 ± 1 and 6037 ± 1 *m/z*, specific to serovar Enteritidis, was present in all 55 isolates, and other serovar-specific mass peaks were absent in these isolates. The mass peak at 7184 ± 1 *m/z*, specific to serovar Typhimurium, was present in 74 isolates. In these strains, the specific peaks for Enteritidis and Thompson were absent. The mass peak at 4925 ± 1 *m/z* was present in all isolates identified as serovar Thompson, whereas this mass peak was absent in serovar Enteritidis and Typhimurium. Therefore, the three mass peaks were considered to be specific peaks that could be used to detect the serovars Enteritidis, Typhimurium, and Thompson.

### 3.4. Serological Identification of Isolates through Agglutination of Antisera

A total of 132 isolates were reidentified using traditional serotyping slide agglutination tests, and the results were compared with serotyping using serovar-specific peaks (Table 5). According to the White–Kauffmann–Le Minor scheme, 55 isolates were determined to be serovar Enteritidis (1,9,12:g,m:-), and all of these isolates were confirmed as serovar Typhimurium by peak analysis. Additionally, 74 isolates were determined as serovar Typhimurium (1,4,[5],12:i:1,2) and 3 as Thompson (6,7,14:k:1,5). The identification of all isolates was consistent with the results of serotyping through the agglutination of antisera and peak analysis. The resolution of serotyping, achieved using antiserum agglutination, is comparable to that of MALDI–TOF MS based on peak analysis, but MALDI-TOF MS is the most rapid and convenient method for the detection of the three serovars.

## 4. Discussion

*Salmonella* Enteritidis has not been found in edible eggs, according to the scope of the inspection standards in Korea, but serovars Typhimurium and Thompson have been added to the inspection standards following recent food poisoning cases. In this study, we evaluated the use of MALDI–TOF MS for the rapid detection of epidemiologically important serovars, with an emphasis on three frequently isolated serovars: Enteritidis, Typhimurium, and Thompson.

The reliability of detection by MALDI–TOF MS depends upon the reference spectra in the database [23]. The commercial database had low taxonomic resolution at the level of subspecies or below [15,16]. The reliability of detection can be improved by adding new reference spectra, as reported in previous studies [14,15,23,24]. However, serovars Enteritidis, Typhimurium, and Thompson could not be detected by supplementing the database since their mass patterns are indistinguishable from other serovars such as Agona, Anatum, and Choleraesuis. However, peak-specific analysis allowed the distinction of the three serovars. This approach, based on the identification of specific protein peaks produced by MALDI–TOF MS, has been used in previous studies to improve the identification rate of different subtypes of the same species, such as *Listeria monocytogenes*, *Vibrio parahaemolyticus*, *Bacillus cereus*, and *Streptococcus* species [20,25,26,27,28]. However, MALDI–TOF–MS-based subtyping for applications such as the identification of serovars, antibiotic resistance strains, and clonal complexes is still challenging and requires careful data analysis [23].

The ability of this approach to discriminate between serovars was evaluated by analyzing multiple mass spectra obtained from 65 specimens representing 56 different serovars. Mass signals at 3018 ± 1, 6037 ± 1, 7184 ± 1, and 4925 ± 1 *m/z* were unique to serovars Enteritidis, Typhimurium, and Thompson and are, therefore, useful for detecting these serovars. The specific mass peak for the identification of Enteritidis (6037 ± 1 *m/z*) was found to be consistent with previous studies, whereas the specific peak of Thompson had never been investigated [7,23]. The specific mass peak for Typhimurium/I 4,[5],12:i:- had also been found in a previous study. *Salmonella* Typhimurium and I 4,[5],12:i:- share similar antigenic formulas, the only difference being one flagellar antigen (1,4,[5],12:i:1,2 vs 4,[5],12:i:-). However, the specific peaks of Typhimurium are not present in other serovars, including I 4,[5],12:i:-, so the mass peak found in the previous study was not considered to be a specific peak in this study. Thus, like Thompson, mass signals at 3018 ± 1 *m/z* for Enteritidis and 7184 ± 1 *m/z* for Typhimurium were first discovered in this study. Of the two mass peaks for Enteritidis, the mass peak at 6037 ± 1 *m/z* was found to be 88% and 93% accurate in previous studies; some strains lacked diagnostic marker ions for Enteritidis [7,23]. A variable expression rate of a specific protein might be observed due to differential regulation, which can lead to false-negatives when the protein peak concentration is below the detection limit [7,23]. However, when fresh isolates were analyzed, the absence of a peak was rarely observed, so the number of false-negatives should be small. The mass peak was barely detectable in some strains when the cell was collected from a single colony instead of from bacterial smears from the same agar plate [7,23]. Unlike previous studies, we found that both peaks for Enteritidis were 100% present in 61 reference strains and isolates of Enteritidis. A fresh colony was used for the analysis, so all of the mass peaks may have been detected in all strains. Additionally, the reproducibility of mass spectra could be affected by strain variability [20]. The mass spectra of a large number of isolates for Enteritidis and Typhimurium were further analyzed to confirm that serovar-specific peaks were consistently present in each serovar. Therefore, the mass peaks for Enteritidis and Typhimurium discovered in this study can be considered to be reliable. However, since Thompson was confirmed using only a limited number of isolates, several more isolates might be required.

Many previous studies have reported that MALDI–TOF MS is rapid and cost-efficient compared to other typing methods [7]. MALDI–TOF MS can be used to detect and identify 10 isolated strains within 15 min from colony selection to final results. The higher the throughput rate of samples that need to be analyzed, the lower the cost of analysis per isolate [7]. MALDI–TOF MS does not exceed USD 0.2 per isolate to detect microorganisms, whereas other detection methods, such as PCR-based methods, cost at least USD 10 per isolate [29,30]. Serotyping by MALDI–TOF MS has a cost at least three times lower than that of other serotyping and biochemical tests [7]. In this study, we propose a serotyping method for the rapid detection of serovars Enteritidis, Typhimurium, and Thompson from serovar-specific mass peaks obtained using MALDI–TOF MS. This serotyping method does not require serovar-specific reagents or manipulations, such as sequence amplification and DNA extraction, and can, therefore, reduce the time, cost, and labor required for *Salmonella* serotyping [23]. The direct transfer method was used to minimize the time required for protein extraction since it is the easiest, cheapest, and fastest way of sample preparation and does not require trained staff [23]. Serotyping using protein mass peaks can reduce the number of samples that need to be analyzed, compared with conventional serotyping and biochemical testing. The specific mass peaks discovered in this study successfully detected serovars Enteritidis, Typhimurium, and Thompson, so the results obtained using these peaks are more accurate and efficient than traditional serotyping methods, which can produce ambiguous results. However, this method has a limited aspect, in that it cannot be automated or used without expert knowledge of MALDI–TOF MS equipment and software.

## 5. Conclusions

MALDI–TOF MS based on mass peaks has proven to be a rapid and convenient method for the detection of *Salmonella* Enteritidis, Typhimurium, and Thompson. The mass peaks found in this study were specific to the three important serovars and were successfully applied to many isolates. Our method will be useful for large-scale, cost-effective screening of serovars and can be applied as an alternative to traditional serotyping methods or as a supplementary method.

## Figures and Tables

**Figure 1 foods-10-00933-f001:**
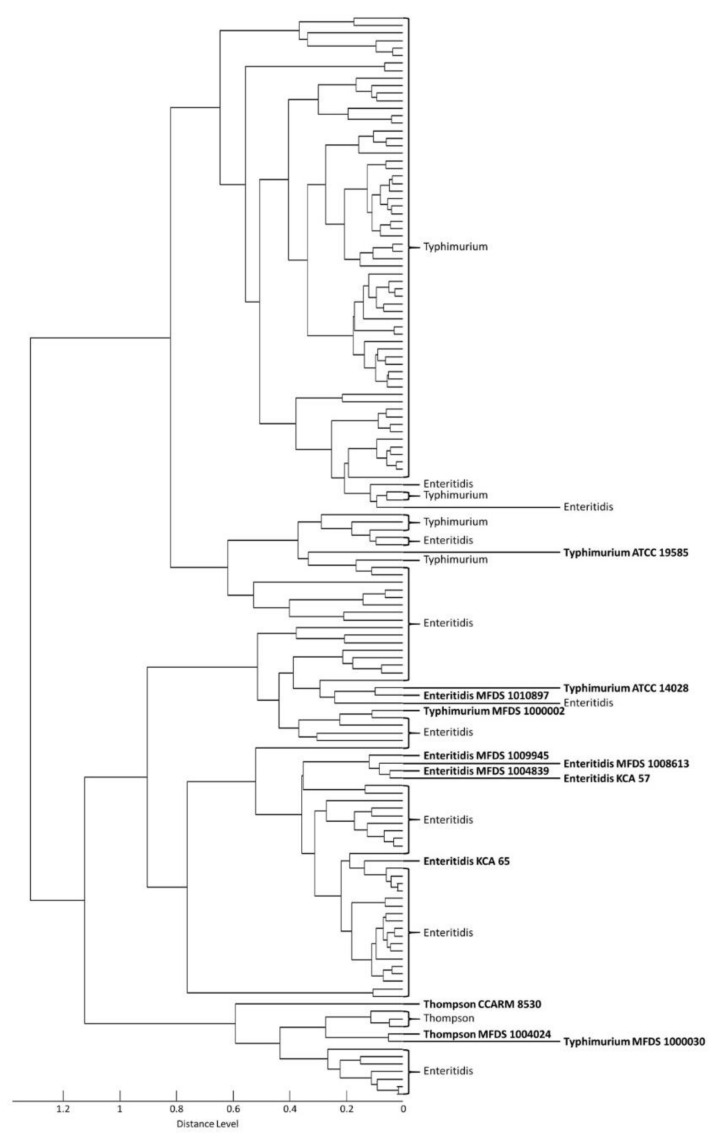
Main spectrum profile (MSP) dendrogram generated based on *m/z* values and relative intensities of 12 reference strains and 132 isolates. Bold indicates reference strains.

**Figure 2 foods-10-00933-f002:**
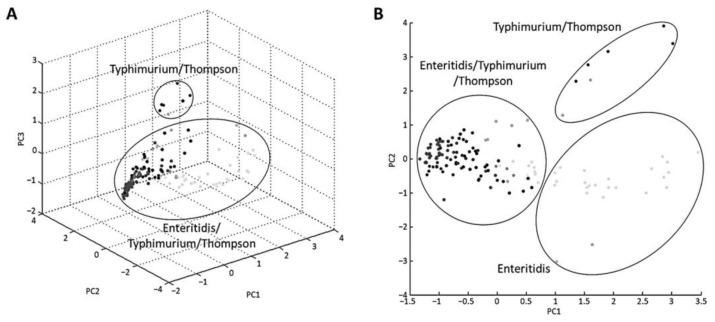
Principal component analysis (PCA) generated by mass spectra of 12 reference strains and 132 isolates. Each dot on the (**A**) three-dimensional plot and (**B**) two-dimensional plot represents strains.

**Figure 3 foods-10-00933-f003:**
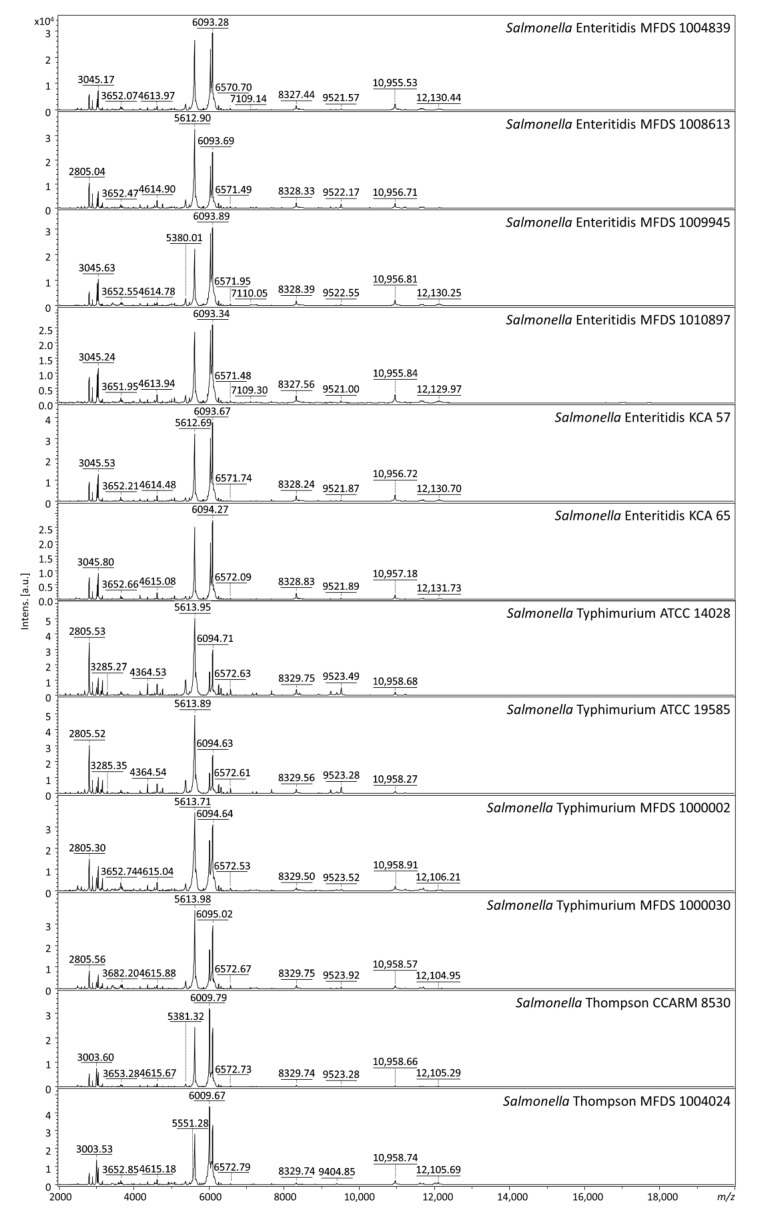
MALDI–TOF MS spectra of reference strains of serovars Enteritidis, Typhimurium, and Thompson; *m/z*, mass-to-charge ratio; a.u., arbitrary units.

**Figure 4 foods-10-00933-f004:**
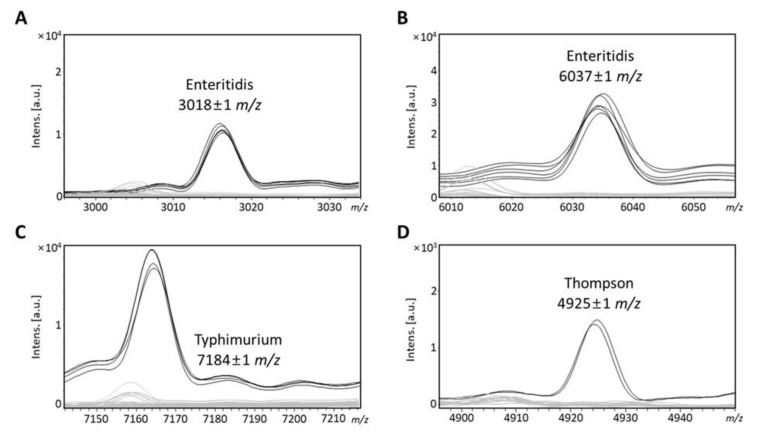
Specific peaks for three serovars. (**A**) Mass peak at 3018 ± 1 *m/z* presents in serovar Enteritidis reference strains, (**B**) mass peak at 6037 ± 1 *m/z* presents in serovar Enteritidis strains, (**C**) mass peak at 7184 ± 1 *m/z* presents in serovar Typhimurium strains, and (**D**) mass peak at 4925 ± 1 *m/z* presents in serovar Thompson.

**Table 1 foods-10-00933-t001:** List of *Salmonella* reference strains used in this study.

Serovars	Designated As:
*Salmonella* Enteritidis	MFDS ^1^ 1004839, MFDS 1008613, MFDS 1009945, MFDS 1010897, KCA ^2^ 57, KCA 65
*Salmonella* Typhimurium	ATCC ^3^ 14028, ATCC 19585, MFDS 1000002, MFDS 1000030
*Salmonella* Thompson	CCARM ^4^ 8530, MFDS 1004024
*Salmonella* Aberdeen	NCCP ^5^ 10142
*Salmonella* Agona	MFDS 1004876
*Salmonella* Albany	MFDS K000014
*Salmonella* Anatum	KVCC ^6^ BA0000586
*Salmonella* Bardo	NCCP 13572
*Salmonella* Bareilly	NCCP 16324
*Salmonella* Berta	KVCC BA0000581
*Salmonella* Blockley	NCCP 10769
*Salmonella* Bovismorbificans	NCCP 12244
*Salmonella* Braenderup	MFDS 1008393
*Salmonella* Brandenburg	NCCP 12835
*Salmonella* Cerro	NCCP 12215
*Salmonella* Choleraesuis	ATCC 13312
*Salmonella* Derby	MFDS 1009813
*Salmonella* Dessau	MFDS 1010078
*Salmonella* Elisabethville	NCCP 14030
*Salmonella* Gallinarum	ATCC 9120
*Salmonella* Give	NCCP 13696
*Salmonella* Hadar	NCCP 13571
*Salmonella* Heidelberg	NCCP 11693
*Salmonella* Hindmarsh	BFR ^7^ 12
*Salmonella* I 4,[5],12:i:-	MFDS 1004858
*Salmonella* Infantis	MFDS 1010567
*Salmonella* Javiana	FDA ^8^ 05
*Salmonella* Kedougou	NCCP 11685
*Salmonella* Kentucky	NCCP 11686
*Salmonella* Kottbus	NCCP 12234
*Salmonella* Litchfield	FDA 33
*Salmonella* Livingstone	MFDS 1004819
*Salmonella* London	MFDS 1004861
*Salmonella* Madelia	FDA 30
*Salmonella* Mbandaka	KVCC BA1800001
*Salmonella* Meleagridis	FDA 34
*Salmonella* Minnesota	MFDS 1008449
*Salmonella* Mississippi	FDA 32
*Salmonella* Montevideo	MFDS 1006814
*Salmonella* Muenchen	KCA 03
*Salmonella* Muenster	FDA 23
*Salmonella* Newington	NCCP 10894
*Salmonella* Newport	MFDS 1005422
*Salmonella* Panama	MFDS 1004857
*Salmonella* Paratyphi A	NCCP 14759
*Salmonella* Paratyphi B	ATCC 10719
*Salmonella* Paratyphi C	ATCC 13428
*Salmonella* Poona	FDA 22
*Salmonella* Reading	MFDS 1007899
*Salmonella* Rissen	NCCP 13709
*Salmonella* Schwarzengrund	MFDS 1006893
*Salmonella* Senftenberg	CCARM 0041
*Salmonella* Singapore	NCCP 12218
*Salmonella* Vinohrady	NCCP 12217
*Salmonella* Virchow	MFDS 1004870
*Salmonella* Weltevreden	NCCP 12239

^1^ MFDS, Ministry of Food & Drug Safety; ^2^ KCPB, Korea Consumer Protection Board; ^3^ ATCC, American Type Culture Collection; ^4^ CCARM, Culture Collection of Antibiotics Resistant Microbes; ^5^ NCCP, National Culture Collection for Pathogens; ^6^ KVCC, Korea Veterinary Culture Collection; ^7^ BFR, Federal Institute for Risk Assessment; ^8^ FDA, US Food and Drug Administration.

**Table 2 foods-10-00933-t002:** List of isolates identified as *Salmonella* species by MALDI–TOF MS.

Source of Sample	Strains (Number of Isolates)
Processed food	S1-S16, S23-S24, S56-S67, S88 (*n* = 31)
Fresh food	S27, S37-S49, S76, S89, S126-S127, S132 (*n* = 19)
Chicken meat	S30-S36, S50-S55, S90-S121 (*n* = 45)
Vegetable	S68-S75, S77-S86 (*n* = 18)
Egg white	S130-S131 (*n* = 2)
Livestock	S17-S22, S25-S26, S28-S29, S87, S122-S125, S128-S129 (*n* = 17)

**Table 3 foods-10-00933-t003:** Presence/absence of specific peaks for 12 reference strains.

Source of Sample	Mass Peak (*m/z*) ^1^			
	3018 ± 1	6037 ± 1	7184 ± 1	4925 ± 1
Enteritidis MFDS 1004839	+	+	-	-
Enteritidis MFDS 1008613	+	+	-	-
Enteritidis MFDS 1009945	+	+	-	-
Enteritidis MFDS 1010897	+	+	-	-
Enteritidis KCA 57	+	+	-	-
Enteritidis KCA 65	+	+	-	-
Typhimurium ATCC 14028	-	-	+	-
Typhimurium ATCC 19585	-	-	+	-
Typhimurium MFDS 1000002	-	-	+	-
Typhimurium MFDS 1000030	-	-	+	-
Thompson CCARM 8530	-	-	-	+
Thompson MFDS 1004024	-	-	-	+

^1^, presence of peak; -, absence of peak.

**Table 4 foods-10-00933-t004:** Presence/absence of specific peaks for 20 other pathogenic bacterial strains.

Strains	Mass Peak (*m/z*)			
	3018 ± 1	6037 ± 1	7184 ± 1	4925 ± 1
*Escherichia coli* NCCP 14039	-	-	-	-
*Escherichia coli* NCCP 14037	-	-	-	-
*Escherichia coli* NCCP 14033	-	-	-	-
*Escherichia coli* MFDS 0064	-	-	-	-
*Escherichia coli* MFDS 5919	-	-	-	-
*Escherichia coli* O157:H7 MFDS 43894	-	-	-	-
*Escherichia coli* O157:H7 ATCC 43890	-	-	-	-
*Staphylococcus aureus* KCTC ^1^ 12113	-	-	-	-
*Staphylococcus aureus* KCTC 1928	-	-	-	-
*Staphylococcus aureus* ATCC 25923	-	-	-	-
*Bacillus cereus* ATCC 10876	-	-	-	-
*Bacillus cereus* KCCM ^2^ 1174	-	-	-	-
*Listeria monocytogenes* KCTC 3569	-	-	-	-
*Listeria ivanovii* ATCC 19119	-	-	-	-
*Vibrio cholerae* ATCC 14033	-	-	-	-
*Vibrio cholerae* ATCC 14035	-	-	-	-
*Shigella flexneri* KCTC 2517	-	-	-	-
*Shigella sonnei* KCTC 2518	-	-	-	-
*Cronobacter sakazakii* ATCC 29544	-	-	-	-
*Citrobacter freundii* ATCC 8090	-	-	-	-

^1^ KCTC, Korean Collection for Type Cultures; ^2^ KCCM, Korean Culture Center of Microorganisms.

**Table 5 foods-10-00933-t005:** Identification of isolates by antisera agglutination and specific peaks.

Strains	Source	Serovars ^1^	MALDI-TOF MS	
			BioTyper	Specific Peak
S1	Processed food	Enteritidis	*Salmonella* sp.	Enteritidis (3017.9, 6038.4 *m/z*)
S2	Processed food	Enteritidis	*Salmonella* sp.	Enteritidis (3017.9, 6038.0 *m/z*)
S3	Processed food	Enteritidis	*Salmonella* sp.	Enteritidis (3017.9, 6038.3 *m/z*)
S4	Processed food	Enteritidis	*Salmonella* sp.	Enteritidis (3017.9, 6038.5 *m/z*)
S5	Processed food	Enteritidis	*Salmonella* sp.	Enteritidis (3017.9, 6038.3 *m/z*)
S6	Processed food	Enteritidis	*Salmonella* sp.	Enteritidis (3017.8, 6038.6 *m/z*)
S7	Processed food	Enteritidis	*Salmonella* sp.	Enteritidis (3018.0, 6038.2 *m/z*)
S8	Processed food	Enteritidis	*Salmonella* sp.	Enteritidis (3018.3, 6038.2 *m/z*)
S9	Processed food	Enteritidis	*Salmonella* sp.	Enteritidis (3018.2, 6038.6 *m/z*)
S10	Processed food	Enteritidis	*Salmonella* sp.	Enteritidis (3018.5, 6039.2 *m/z*)
S11	Processed food	Enteritidis	*Salmonella* sp.	Enteritidis (3018.0, 6038.3 *m/z*)
S12	Processed food	Enteritidis	*Salmonella* sp.	Enteritidis (3018.3, 6038.8 *m/z*)
S13	Processed food	Enteritidis	*Salmonella* sp.	Enteritidis (3017.5, 6037.5 *m/z*)
S14	Processed food	Enteritidis	*Salmonella* sp.	Enteritidis (3017.7, 6037.7 *m/z*)
S15	Processed food	Enteritidis	*Salmonella* sp.	Enteritidis (3018.0, 6038.5 *m/z*)
S16	Processed food	Enteritidis	*Salmonella* sp.	Enteritidis (3017.9, 6038.1 *m/z*)
S17	Livestock	Enteritidis	*Salmonella* sp.	Enteritidis (3017.9, 6038.0 *m/z*)
S18	Livestock	Enteritidis	*Salmonella* sp.	Enteritidis (3017.9, 6038.1 *m/z*)
S19	Livestock	Enteritidis	*Salmonella* sp.	Enteritidis (3018.1, 6038.6 *m/z*)
S20	Livestock	Enteritidis	*Salmonella* sp.	Enteritidis (3017.9, 6038.3 *m/z*)
S21	Livestock	Enteritidis	*Salmonella* sp.	Enteritidis (3018.0, 6038.3 *m/z*)
S22	Livestock	Enteritidis	*Salmonella* sp.	Enteritidis (3018.2, 6038.5 *m/z*)
S23	Processed food	Enteritidis	*Salmonella* sp.	Enteritidis (3018.1, 6038.4 *m/z*)
S24	Processed food	Enteritidis	*Salmonella* sp.	Enteritidis (3018.0, 6038.2 *m/z*)
S25	Livestock	Enteritidis	*Salmonella* sp.	Enteritidis (3017.7, 6037.4 *m/z*)
S26	Livestock	Enteritidis	*Salmonella* sp.	Enteritidis (3017.7, 6037.6 *m/z*)
S27	Fresh food	Enteritidis	*Salmonella* sp.	Enteritidis (3017.8, 6038.0 *m/z*)
S28	Livestock	Enteritidis	*Salmonella* sp.	Enteritidis (3017.8, 6038.1 *m/z*)
S29	Livestock	Enteritidis	*Salmonella* sp.	Enteritidis (3018.0, 6038.2 *m/z*)
S30	Chicken meat	Enteritidis	*Salmonella* sp.	Enteritidis (3018.0, 6038.7 *m/z*)
S31	Chicken meat	Enteritidis	*Salmonella* sp.	Enteritidis (3017.9, 6038.3 *m/z*)
S32	Chicken meat	Enteritidis	*Salmonella* sp.	Enteritidis (3018.0, 6038.2 *m/z*)
S33	Chicken meat	Enteritidis	*Salmonella* sp.	Enteritidis (3018.8, 6038.4 *m/z*)
S34	Chicken meat	Enteritidis	*Salmonella* sp.	Enteritidis (3018.1, 6038.8 *m/z*)
S35	Chicken meat	Enteritidis	*Salmonella* sp.	Enteritidis (3018.7, 6037.9 *m/z*)
S36	Chicken meat	Enteritidis	*Salmonella* sp.	Enteritidis (3017.9, 6038.1 *m/z*)
S37	Fresh food	Enteritidis	*Salmonella* sp.	Enteritidis (3018.0, 6038.5 *m/z*)
S38	Fresh food	Enteritidis	*Salmonella* sp.	Enteritidis (3018.1, 6038.6 *m/z*)
S39	Fresh food	Enteritidis	*Salmonella* sp.	Enteritidis (3018.1, 6038.8 *m/z*)
S40	Fresh food	Enteritidis	*Salmonella* sp.	Enteritidis (3018.0, 6038.5 *m/z*)
S41	Fresh food	Enteritidis	*Salmonella* sp.	Enteritidis (3018.1, 6038.9 *m/z*)
S42	Fresh food	Enteritidis	*Salmonella* sp.	Enteritidis (3018.2, 6038.8 *m/z*)
S43	Fresh food	Enteritidis	*Salmonella* sp.	Enteritidis (3017.6, 6037.5 *m/z*)
S44	Fresh food	Enteritidis	*Salmonella* sp.	Enteritidis (3017.8, 6038.0 *m/z*)
S45	Fresh food	Enteritidis	*Salmonella* sp.	Enteritidis (3018.0, 6038.3 *m/z*)
S46	Fresh food	Enteritidis	*Salmonella* sp.	Enteritidis (3017.9, 6038.0 *m/z*)
S47	Fresh food	Enteritidis	*Salmonella* sp.	Enteritidis (3017.9, 6038.2 *m/z*)
S48	Fresh food	Enteritidis	*Salmonella* sp.	Enteritidis (3018.1, 6038.4 *m/z*)
S49	Fresh food	Enteritidis	*Salmonella* sp.	Enteritidis (3018.3, 6038.8 *m/z*)
S50	Chicken meat	Enteritidis	*Salmonella* sp.	Enteritidis (3018.0, 6038.4 *m/z*)
S51	Chicken meat	Enteritidis	*Salmonella* sp.	Enteritidis (3018.2, 6037.6 *m/z*)
S52	Chicken meat	Enteritidis	*Salmonella* sp.	Enteritidis (3018.6, 6037.8 *m/z*)
S53	Chicken meat	Enteritidis	*Salmonella* sp.	Enteritidis (3018.7, 6038.6 *m/z*)
S54	Chicken meat	Enteritidis	*Salmonella* sp.	Enteritidis (3019.0, 6040.4 *m/z*)
S55	Chicken meat	Enteritidis	*Salmonella* sp.	Enteritidis (3019.0, 6038.7 *m/z*)
S56	Processed food	Typhimurium	*Salmonella* sp.	Typhimurium (7183.2 *m/z*)
S57	Processed food	Typhimurium	*Salmonella* sp.	Typhimurium (7183.5 *m/z*)
S58	Processed food	Typhimurium	*Salmonella* sp.	Typhimurium (7183.5 *m/z*)
S59	Processed food	Typhimurium	*Salmonella* sp.	Typhimurium (7183.4 *m/z*)
S60	Processed food	Typhimurium	*Salmonella* sp.	Typhimurium (7183.2 *m/z*)
S61	Processed food	Typhimurium	*Salmonella* sp.	Typhimurium (7183.2 *m/z*)
S62	Processed food	Typhimurium	*Salmonella* sp.	Typhimurium (7183.8 *m/z*)
S63	Processed food	Typhimurium	*Salmonella* sp.	Typhimurium (7183.5 *m/z*)
S64	Processed food	Typhimurium	*Salmonella* sp.	Typhimurium (7183.1 *m/z*)
S65	Processed food	Typhimurium	*Salmonella* sp.	Typhimurium (7183.4 *m/z*)
S66	Processed food	Typhimurium	*Salmonella* sp.	Typhimurium (7183.7 *m/z*)
S67	Processed food	Typhimurium	*Salmonella* sp.	Typhimurium (7184.1 *m/z*)
S68	Vegetable	Typhimurium	*Salmonella* sp.	Typhimurium (7184.2 *m/z*)
S69	Vegetable	Typhimurium	*Salmonella* sp.	Typhimurium (7184.5 *m/z*)
S70	Vegetable	Typhimurium	*Salmonella* sp.	Typhimurium (7184.4 *m/z*)
S71	Vegetable	Typhimurium	*Salmonella* sp.	Typhimurium (7184.8 *m/z*)
S72	Vegetable	Typhimurium	*Salmonella* sp.	Typhimurium (7185.0 *m/z*)
S73	Vegetable	Typhimurium	*Salmonella* sp.	Typhimurium (7183.2 *m/z*)
S74	Vegetable	Typhimurium	*Salmonella* sp.	Typhimurium (7183.7 *m/z*)
S75	Vegetable	Typhimurium	*Salmonella* sp.	Typhimurium (7184.2 *m/z*)
S76	Fresh food	Typhimurium	*Salmonella* sp.	Typhimurium (7184.1 *m/z*)
S77	Vegetable	Typhimurium	*Salmonella* sp.	Typhimurium (7184.3 *m/z*)
S78	Vegetable	Typhimurium	*Salmonella* sp.	Typhimurium (7183.1 *m/z*)
S79	Vegetable	Typhimurium	*Salmonella* sp.	Typhimurium (7183.3 *m/z*)
S80	Vegetable	Typhimurium	*Salmonella* sp.	Typhimurium (7183.3 *m/z*)
S81	Vegetable	Typhimurium	*Salmonella* sp.	Typhimurium (7183.1 *m/z*)
S82	Vegetable	Typhimurium	*Salmonella* sp.	Typhimurium (7183.8 *m/z*)
S83	Vegetable	Typhimurium	*Salmonella* sp.	Typhimurium (7183.2 *m/z*)
S84	Vegetable	Typhimurium	*Salmonella* sp.	Typhimurium (7183.3 *m/z*)
S85	Vegetable	Typhimurium	*Salmonella* sp.	Typhimurium (7184.0 *m/z*)
S86	Vegetable	Typhimurium	*Salmonella* sp.	Typhimurium (7184.1 *m/z*)
S87	Livestock	Typhimurium	*Salmonella* sp.	Typhimurium (7184.3 *m/z*)
S88	Processed food	Typhimurium	*Salmonella* sp.	Typhimurium (7184.8 *m/z*)
S89	Fresh food	Typhimurium	*Salmonella* sp.	Typhimurium (7184.4 *m/z*)
S90	Chicken meat	Typhimurium	*Salmonella* sp.	Typhimurium (7184.5 *m/z*)
S91	Chicken meat	Typhimurium	*Salmonella* sp.	Typhimurium (7184.6 *m/z*)
S92	Chicken meat	Typhimurium	*Salmonella* sp.	Typhimurium (7185.0 *m/z*)
S93	Chicken meat	Typhimurium	*Salmonella* sp.	Typhimurium (7184.6 *m/z*)
S94	Chicken meat	Typhimurium	*Salmonella* sp.	Typhimurium (7183.3 *m/z*)
S95	Chicken meat	Typhimurium	*Salmonella* sp.	Typhimurium (7183.9 *m/z*)
S96	Chicken meat	Typhimurium	*Salmonella* sp.	Typhimurium (7183.9 *m/z*)
S97	Chicken meat	Typhimurium	*Salmonella* sp.	Typhimurium (7184.3 *m/z*)
S98	Chicken meat	Typhimurium	*Salmonella* sp.	Typhimurium (7184.1 *m/z*)
S99	Chicken meat	Typhimurium	*Salmonella* sp.	Typhimurium (7184.8 *m/z*)
S100	Chicken meat	Typhimurium	*Salmonella* sp.	Typhimurium (7185.4 *m/z*)
S101	Chicken meat	Typhimurium	*Salmonella* sp.	Typhimurium (7184.0 *m/z*)
S102	Chicken meat	Typhimurium	*Salmonella* sp.	Typhimurium (7184.1 *m/z*)
S103	Chicken meat	Typhimurium	*Salmonella* sp.	Typhimurium (7184.1 *m/z*)
S104	Chicken meat	Typhimurium	*Salmonella* sp.	Typhimurium (7183.3 *m/z*)
S105	Chicken meat	Typhimurium	*Salmonella* sp.	Typhimurium (7183.9 *m/z*)
S106	Chicken meat	Typhimurium	*Salmonella* sp.	Typhimurium (7183.7 *m/z*)
S107	Chicken meat	Typhimurium	*Salmonella* sp.	Typhimurium (7184.1 *m/z*)
S108	Chicken meat	Typhimurium	*Salmonella* sp.	Typhimurium (7184.1 *m/z*)
S109	Chicken meat	Typhimurium	*Salmonella* sp.	Typhimurium (7184.1 *m/z*)
S110	Chicken meat	Typhimurium	*Salmonella* sp.	Typhimurium (7184.8 *m/z*)
S111	Chicken meat	Typhimurium	*Salmonella* sp.	Typhimurium (7184.4 *m/z*)
S112	Chicken meat	Typhimurium	*Salmonella* sp.	Typhimurium (7184.8 *m/z*)
S113	Chicken meat	Typhimurium	*Salmonella* sp.	Typhimurium (7183.1 *m/z*)
S114	Chicken meat	Typhimurium	*Salmonella* sp.	Typhimurium (7183.4 *m/z*)
S115	Chicken meat	Typhimurium	*Salmonella* sp.	Typhimurium (7183.4 *m/z*)
S116	Chicken meat	Typhimurium	*Salmonella* sp.	Typhimurium (7183.3 *m/z*)
S117	Chicken meat	Typhimurium	*Salmonella* sp.	Typhimurium (7184.1 *m/z*)
S118	Chicken meat	Typhimurium	*Salmonella* sp.	Typhimurium (7184.6 *m/z*)
S119	Chicken meat	Typhimurium	*Salmonella* sp.	Typhimurium (7184.3 *m/z*)
S120	Chicken meat	Typhimurium	*Salmonella* sp.	Typhimurium (7184.8 *m/z*)
S121	Chicken meat	Typhimurium	*Salmonella* sp.	Typhimurium (7184.9 *m/z*)
S122	Livestock	Typhimurium	*Salmonella* sp.	Typhimurium (7183.2 *m/z*)
S123	Livestock	Typhimurium	*Salmonella* sp.	Typhimurium (7184.6 *m/z*)
S124	Livestock	Typhimurium	*Salmonella* sp.	Typhimurium (7183.3 *m/z*)
S125	Livestock	Typhimurium	*Salmonella* sp.	Typhimurium (7184.9 *m/z*)
S126	Fresh food	Typhimurium	*Salmonella* sp.	Typhimurium (7175.0 *m/z*)
S127	Fresh food	Typhimurium	*Salmonella* sp.	Typhimurium (7184.6 *m/z*)
S128	Livestock	Typhimurium	*Salmonella* sp.	Typhimurium (7184.9 *m/z*)
S129	Livestock	Typhimurium	*Salmonella* sp.	Typhimurium (7184.7 *m/z*)
S130	Egg white	Thompson	*Salmonella* sp.	Thompson (4925.1 *m/z*)
S131	Egg white	Thompson	*Salmonella* sp.	Thompson (4924.8 *m/z*)
S132	Fresh food	Thompson	*Salmonella* sp.	Thompson (4925.0 *m/z*)

^1^ Determined serovars through antisera agglutination.

## Data Availability

The data presented in this study are available on request from the corresponding author.

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
