# Peer review of "Rapid Detection of Salmonella Enteritidis, Typhimurium, and Thompson by Specific Peak Analysis Using Matrix-Assisted Laser Desorption Ionization Time-of-Flight Mass Spectrometry"

_foods, 2021, doi:10.3390/foods10050933_

Round 1

Reviewer 1 Report

General comments

Very interesting well-written study.

Please, put italics in enteritidis, typhimurium and Thompson and generally all species in all manuscript.

Line 25-26: Cause and result. Please rephrase the sentence.

Line 35: …..and even lead to death….

Line 80: I suppose strains and not stains

Line 113-114: How many poor-quality spectra were excluded from the study? If it was a significant number, you should explain more on this issue?

Line 124: You refer 4 categories and explain only three?

Figure 4C: There is no mass peak at 7,184m/z. Please explain.

Line 249: 65 specimens representing 55 different serovars. In a previous line you mentioned 65 strains into 56 different serovars. Please explain.

Can the source of isolation affect the final results?

Finally, are there other bacteria which can give the same mass peaks and therefore mislead your results. Please explain

Author Response

Response to Reviewer 1 Comments

1. Please, put italics in enteritidis, typhimurium and Thompson and generally all species in all manuscript.

Response: According to the system of Salmonella nomenclature sed by WHO and CDC, the serovar name is not italicized and starts with a capital letter (Brenner et al., J. Clin. Microbiol., 2000). Therefore, to avoid confusion between serovars and species, many researchers have referred to a serovar name that is not italicized and has the first letter capitalized. Likewise, in this study, serovar names were indicated in capital letters and non-italic letters, such as Enteritidis, Typhimurium, and Thompson.

2. Line 25-26: Cause and result. Please rephrase the sentence.

Response: As you recommended, we rephrased the sentence in lines 25-26 as follows:

Lines 25-26: Our method can rapidly detect a large number of strains, so it would be useful for the high-throughput screening of Salmonella serovars.

3. Line 35: ….and even lead to death….

Response: As you recommended, we revised the sentence in line 36 as follows:

Line 36: infection occur annually in the United States alone, and even lead to death, and Salmonella

4. Line 80: I suppose strains and not stains

Response: As you recommended, we corrected the sentence in line 81 as follows:

Line 81: Salmonella strains

5. Line 113-114: How many poor-quality spectra were excluded from the study? If it was a significant number, you should explain more on this issue?

Response: Only one peak (intensity value 36.189 arbitrary unit) was excluded due to poor quality spectra with low intensity from this study. As you recommended, we added the sentence in lines 181-182 as follows:

Lines 181-182: In this process, one peak was excluded due to poor quality spectra with low intensity (36.189 arbitrary unit).

6. Line 124: You refer 4 categories and explain only three?

Response: As you recommended, we revised the sentence in lines 124-128 as follows:

Lines 124-128: The score interpretation of the four categories was as follows: a score of ≥ 2.3 indicated a high level of probability of species, scores between 2.000–2.299 indicates probable species identification, scores between 1.700–1.999 indicates probable genus identification, and a score <1.7 indicates no reliable identification.

7. Figure 4C: There is no mass peak at 7,184 m/z. Please explain.

Response: The intensity of Typhimurium specific mass peak was relatively lower than the left peak (7,165 m/z). In the peak analysis, a mass peak at 7,184 m/z was detected in all Typhimurium strains with an intensity ranging from 1281.269 to 5305.206 arbitrary unit. We adjusted the x- and y-axis scale in Figure 4C.

8. Line 249: 65 specimens representing 55 different serovars. In a previous line you mentioned 65 strains into 56 different serovars. Please explain.

Response: As you recommended, we revised the sentence in line 257 as follows:

Line 257: 65 specimens representing 56 different serovars.

9. Can the source of isolation affect the final results?

Response: In this study, each specific peak existed in all isolates regardless of the source of isolation. Also, in the MSP-dendrogram and PCA clustering, a difference in mass peak pattern according to the source of isolation was not observed. We added the sentence in lines 163-164 as follows:

Lines 163-164: Also, a difference in mass peak pattern according to the source of isolation was not observed.

10. Finally, are there other bacteria which can give the same mass peaks and therefore mislead your results. Please explain

Response: As you recommended, we performed an additional analysis of the presence or absence of the specific mass peaks in other pathogenic bacteria. As a result, specific mass peaks do not exist in other pathogenic bacterial strains. We added the sentence about the analysis result in lines 191-193 as follows:

Lines 191-193: Mass peaks at 3,018 ± 1, 6,037 ± 1, 7,184 ± 1, and 4,925 ± 1 m/z were unique to serovars Enteritidis, Typhimurium, and Thompson, and 20 other pathogenic bacterial strains (Table 4).

And we newly added the presence or absence of specific mass peaks in other bacterial strains in Table 4.

Reviewer 2 Report

The article entitled “Rapid detection of Salmonella Enteritidis, Typhimurium, and Thompson by specific peak analysis using matrix-assisted laser desorption ionization time-of-flight mass spectrometry” is well written and explores the use of MALDI-TOF-MS in characterizing Salmonella enterica sbsps enterica to some Serovars associated with food contamination. Unfortunately, although some differences were found, they could not be automated or use without expert knowledge on the equipment and Salmonella serovars. Therefore, limiting their findings. Furthermore, only a limited number of Salmonella serovars were used for this study, and we do now know if the minor differences found could be present in other serovars. Therefore, although the information presented here has some academic value, the goal of expediting the typing of this zoonotic pathogen is limited by its limited use and likely associated with a particular brand of equipment. The authors should limit their enthusiasm for their findings and mention the limiting aspects of their work.

Author Response

Response to Reviewer 2 Comments

The article entitled “Rapid detection of Salmonella Enteritidis, Typhimurium, and Thompson by specific peak analysis using matrix-assisted laser desorption ionization time-of-flight mass spectrometry” is well written and explores the use of MALDI-TOF-MS in characterizing Salmonella enterica subsp enterica to some Serovars associated with food contamination. Unfortunately, although some differences were found, they could not be automated or use without expert knowledge on the equipment and Salmonella serovars. Therefore, limiting their findings. Furthermore, only a limited number of Salmonella serovars were used for this study, and we do now know if the minor differences found could be present in other serovars. Therefore, although the information presented here has some academic value, the goal of expediting the typing of this zoonotic pathogen is limited by its limited use and likely associated with a particular brand of equipment. The authors should limit their enthusiasm for their findings and mention the limiting aspects of their work.

Response: The purpose of this study is to rapidly detect three epidemiologically important serovars in Korea among the many serovars. And, in the recent study, 11 other serovars were used to verify the specific peaks for five predominant serovars in a Thai broiler industry, whereas in this study, 56 other serovars, including mainly prevalent serovars, were used for specific mass peaks validation (Mangmee et al., Food Control, 2020). Therefore, it cannot be a limiting aspect that only a limited number of Salmonella serovars were used for this study. However, we agree that this study has limitations in that it cannot be automated or use without expert knowledge on the equipment. As you recommended, we added the sentence that mentions the limiting aspects of this work in lines 305-307 as follows:

Lines 305-307: However, this method has a limited aspect that it could not be automated or use without expert knowledge on the MALDI-TOF MS equipment and software.

Round 2

Reviewer 1 Report

My comments have been addressed. Thanks!